# Epidemiology, injury pattern and outcome of older trauma patients: A 15-year study of level-I trauma centers

Axel Benhamed[1,2,3,4]☯, Brice Batomen[1,5]☯, Valérie Boucher[1]‡, Krishan Yadav[6], Éric Mercier[1,2], Chartelin Jean Isaac[1]‡, Mélanie Bérubé[1,7], Francis Bernard[8], Jean-Marc Chauny[9], Lynne Moore[10], Marie Josée Sirois[1], Karim Tazarourte[3,4], Amaury Gossiome[1,3], Marcel Émond[1,2]☯*

1 Centre de recherche du CHU de Québec-Université Laval, Québec, QC, Canada, 2 Département de médecine d'urgence, CHU de Québec-Université Laval, Québec, QC, Canada, 3 Hospices Civils de Lyon, Service d'Accueil des Urgences–SAMU 69, Centre Hospitalier Universitaire Édouard Herriot, Lyon, France, 4 Research On Healthcare Performance (RESHAPE), INSERM U1290, Université Claude Bernard Lyon 1, Lyon, France, 5 Dalla Lana school of public health, University of Toronto, Toronto, Ontario, Canada, 6 Department of Emergency Medicine, University of Ottawa, Ottawa, Ontario, Canada, 7 Faculty of Nursing, Université Laval, Québec, QC, Canada, 8 Section of Critical Care, Department of Medicine, University of Montreal, Montreal, Canada, 9 Department of Emergency Medicine, Research Center, CIUSSS-Nord-de-l'Île de-Montréal, Hôpital Sacré-Cœur de Montréal, Montréal, QC, Canada, 10 Department of Social and Preventative Medicine, Université Laval, Québec City, QC, Canada

☯ These authors contributed equally to this work.
‡ VB and CJI also contributed equally to this work.
* marcel.emond@fmed.ulaval.ca

## Abstract

### Background

Older adults have become a significant portion of the trauma population. Exploring their specificities is crucial to better meet their specific needs. The primary objective was to evaluate the temporal changes in the incidence, demographic and trauma characteristics, injury pattern, in-hospital admission, complications, and outcome of older trauma patients.

### Methods

A multicenter retrospective cohort study was conducted using the Quebec Trauma Registry. Patients aged ≥16 years admitted to one of the three adult level-I trauma centers between 2003 and 2017 were included. Descriptive analyses and trend-tests were performed to describe temporal changes.

### Results

A total of 53,324 patients were included, and 24,822 were aged ≥65 years. The median [IQR] age increased from 57[36–77] to 67[46–82] years, and the proportion of older adults rose from 41.8% in 2003 to 54.1% in 2017. Among those, falls remain the main mechanism (84.7%-88.3%), and the proportion of severe thorax (+8.9%), head (+8.7%), and spine (+5%) injuries significantly increased over time. The proportion of severely injured older patients almost doubled (17.6%-32.3%), yet their mortality decreased (-1.0%). Their average annual bed-days consumption also increased (+15,004 and +1,437 in non-intensive care wards and ICU, respectively).

**Data Availability Statement:** This study analyzed anonymous data from a provincial trauma registry (Registre des traumatismes du Québec). By

provincial laws, the authors are not at liberty to share the datasets used for their analyses, as they were required to sign a non-disclosure agreement to access this data.Access to the RTQ datasets may be formally requested to the Commission d'Accès à l'information (https://www.cai.gouv.qc.ca) and the Direction de l'analyse et de la gestion de l'information of the Régie de l'assurance maladie du Québec (https://numerique.banq.qc.ca/patrimoine/details/52327/49342). Here is the contact information where information on how to request access to this governmental database: statistiques@ramq.gouv.qc.ca; Protectiondesrenseignementspersonnels@ramq.gouv.qc.ca.

**Funding:** AB received a scholarship from the Fondation du CHU de Québec (no grant number). This project was funded by the Fonds de recherche du Québec – Santé (#33239). These funding agencies were in no way involved in the design, data collection and analyses of this study.

**Competing interests:** The authors have declared that no competing interests exist.

## Conclusions

Since 2014, older adults have represented the majority of admissions in Level-I trauma centers in Québec. Their bed-days consumption has greatly increased, and their injury pattern and severity have deeply evolved, while we showed a decrease in mortality.

## Introduction

Trauma is the leading cause of death among patients aged under 40 years and is, therefore, a significant concern in young adults [1]. However, the increased life expectancy contributes to a worldwide phenomenon: an important shift in the demographics of trauma patients. For instance, patients aged 65 years and over in Australia accounted for a third of all major trauma admissions [2]. In the United Kingdom, the second most represented age group for major trauma patients was 75 years and older [3]. The proportion of older trauma patients in the National Trauma Database (USA) increased from 18% (2005) to 30% (2015), and the mean age of trauma patients rose from 39 (1993) to 51 years (2013) in a German Trauma Register [4]. Canada is no exception; Statistics Canada reported that in 2016, older adults aged 65 years and over (>5.9 million) outnumbered children aged 0–14 years (5.8 million) in the general population [5]. Yet, no data have been published regarding the evolution of Canadian trauma patients' characteristics.

Advanced age has been widely associated with undertriage [6–8], longer in-hospital length of stay (LOS) [9] and poor outcomes such as higher mortality, morbidity and long-term functional deficits [9–11]. Furthermore, this population's physiological changes and pre-existing comorbidities may lead to a loss of reserve capacities to adapt to physiological stressors. Providing dedicated care to older patients who sustained trauma is widely recognized as a significant challenge for clinicians. A better understanding of the characteristics and specificities of these older trauma patients, including their evolution over time, may provide helpful information to adapt healthcare resources to better meet the needs of this growing population. However, the literature on older trauma patients is still scarce.

The primary objective of this study was to evaluate the temporal changes in the incidence, demographics, injury pattern, in-hospital admission, complications, and outcome of older trauma patients (≥65 years). Secondary objective was to compare these changes to those observed in younger patients.

## Methods

### Study design and setting

We conducted a multicenter retrospective cohort study using data from the Quebec Trauma Registry (RTQ). This registry is managed by Quebec's Ministry of Health and Social Services and serves a population of 8.5 million in a geographic area of approximately 1.7 million km$^2$ [12]. It contains information on all trauma patients (International Classification of Diseases codes 800–959) treated at one of the 59 designated trauma centers of the province's inclusive trauma system. All patients admitted to the hospital after a visit to an ED following trauma are included. Trauma level designations are based on the American College of Surgeons' criteria and are periodically revised [13, 14]. Data from this provincial registry is prospectively extracted and coded by trained medical archivists using patient charts [15]. The RTQ is subject to systematic validation to identify and rectify aberrant data values and to verify date and time chronology.

This project was approved by the CHU de Québec-Université Laval Research ethics board and the *Commission d'Accès à l'information;* therefore, patient consent was not required. Data were all anonymous before analysis. Results are reported as per the STROBE guidelines [16].

## Study population

We included patients aged ≥16 years, admitted or transported (from the prehospital setting or interfacility transfer) to one of the three adult level-I trauma centers between March 1st, 2003 and December 31st, 2017. Because no data regarding the trauma mechanism and injury pattern of patients deceased before or upon ED arrival are collected in the RTQ, these patients were excluded from our analyses.

## Measurements and outcomes

The independent variables included age, sex, mechanism of injury, comorbidities, Abbreviated Injury Scale (AIS), hospital length of stay (LOS), defined as the period from ED arrival to hospital discharge, ICU admission, discharge destination and mortality. Mortality was defined as any death occurring between ED arrival and hospital discharge. The Injury Severity Score (ISS) was calculated after anatomic assessments based on the AIS98 version for patients included before 2013 and the AIS08 version for those included between 2013 and 2017. We used the AIS98 to AIS08 conversion to ensure comparability throughout the years [17]. Patients with an ISS>12 were considered severely injured [18]. We also described bed-day consumption, defined as a day during which a patient is confined to a hospital bed where they stay overnight. The sum of each patient's bed-days consumption was calculated each year to estimate the hospital burden in the two age groups. Based on published expert consensus [19], the following conditions were considered as complications: acute respiratory distress syndrome, aspiration pneumonia, acute respiratory failure, nosocomial pneumonia, pulmonary embolism, cardiac arrest, postoperative haemorrhagic shock, myocardial infarction, abdominal compartment syndrome, anastomotic leak, C.difficile colitis, evisceration/dehiscence, coagulopathy, catheter-related bloodstream infection, sepsis/severe sepsis/shock, wound infection, acute kidney failure, decubitus ulcers, extremity compartment syndrome, nonunion fracture, osteomyelitis, stroke, deep vein thrombosis and delirium.

## Statistical analysis

We calculated frequencies and percentages for categorical variables. Continuous non-normally distributed data were reported as median and interquartile range [IQR]. Continuous normally distributed data were reported as means and standard deviations. Patients were stratified into two age groups: 16–64 years and 65 years and older [20]. Categorical variables were compared using Pearson's Chi-square test. We specifically evaluated the proportion of older adults presenting one of those three common injuries: hip fracture, rib fracture, and traumatic brain injury (TBI). We also performed a temporal trend analysis using Cochran-Armitage trend test ("p-trend"). Missing data were not imputed and are reported. Statistical analyses were conducted using SAS (Statistical Analysis System v9.4, SAS Institute Inc., Cary, NC, USA).

## Results

### Characteristics of study subjects

A total of 53,324 patients were included, and those aged ≥65 years accounted for 46.5% of the cohort (n = 24,822). This group included fewer males than those aged 16–64 years (39.9% *vs* 72.9%, p<0.001), more falls (86.5% *vs* 38.2%), fewer motor vehicle collisions (MVC) (9.1% *vs*

37.8%), and the severity of their injuries was lower (ISS 9 [6–14] *vs* 10 [5–20], p<0.001). A total of 35.1% of older adults were admitted to a level-I trauma center after an inter-facility transfer compared to 51.7% of patients aged 16–64 years. The two body regions that most frequently sustained severe injuries (AIS≥3) were lower extremities (35.8% and 21.5%) and head (24.5% and 27.9%) for older and younger patients, respectively. In-hospital mortality was higher in older patients (10% *vs* 3.9%, p<0.001).

## Demographic characteristics

The overall median age increased from 57 [36–77] years in 2003 to 67 [46–82] years in 2017, and a 12.3% increase was observed in the proportion of injured older patients (41.8%-54.1%). Starting in 2014, older patients represent the majority of trauma-related admissions (Fig 1). Furthermore, we noted a significant increase of the proportion of older men over the years (p-trend <0.001) which rose by 9.9% (32.6%-42.5%), while a non-statistically significant increase (p-trend = 0.15) was observed in younger patients (71.2%-74.5%, Table 1).

## Mechanism of injury

Falls remained the main trauma mechanism among older adults over our study period (between 84.7% and 88.3%). The proportions of MVC slightly decreased in both groups (from 9.9% to 7.8% and from 41.6% to 37.8% among older and younger adults, respectively). Gunshot injuries (1.7% and 0.1%) and stab wounds (4.3% and 0.3%) were rare in both groups (Fig 2).

## Injury pattern

Tables 2 and 3 show the proportion of patients who sustained AIS≥1 and AIS≥3 injuries by body region. Both older and younger adult groups showed a significant increased proportion of patients with severe thorax (+8.9% and +15.8%, p-trend<0.001), and spine injuries (+5.0%, p-trend = 0.003 and +5.1%, p-trend = 0.03), while the proportion of upper (-2.8% and -3.7%,

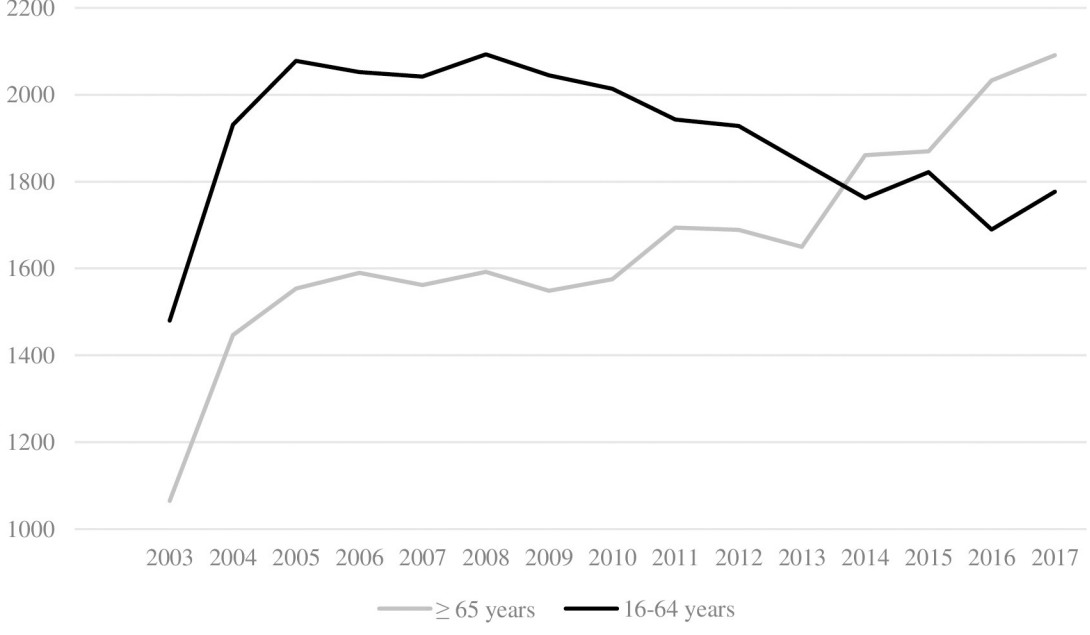

**Fig 1. Annual trauma admissions over the years.**

**Table 1. Demographic characteristics over the years.**

| | | 2003 n = 2,545 | 2004 n = 3,378 | 2005 n = 3,632 | 2006 n = 3,642 | 2007 n = 3,604 | 2008 n = 3,685 | 2009 n = 3,594 | 2010 n = 3,589 | 2011 n = 3,637 | 2012 n = 3,617 | 2013 n = 3,495 | 2014 n = 3,623 | 2015 n = 3,692 | 2016 n = 3,723 | 2017 n = 3,868 | p-trend |
|---|---|---|---|---|---|---|---|---|---|---|---|---|---|---|---|---|---|
| Age, median [IQR] | | 57 [36–77] | 58 [37–78] | 58 [37–79] | 59 [37–78] | 59 [38–79] | 59 [38–78] | 59 [38–78] | 60 [38–78] | 61 [40–79] | 62 [41–79] | 63 [43–79] | 65 [46–81] | 65 [45–81] | 68 [48–82] | 67 [46–82] | <0.001 |
| Sex, male | ≥65 | 347 (32.6) | 473 (32.7) | 530 (34.1) | 542 (34.1) | 602 (38.5) | 621 (39.0) | 620 (40.0) | 629 (39.9) | 715 (42.2) | 701 (41.5) | 724 (43.9) | 806 (43.3) | 844 (45.1) | 865 (42.5) | 888 (42.5) | <0.001 |
| | 16–64 | 1,102 (74.5) | 1,406 (72.8) | 1,532 (73.7) | 1,511 (73.6) | 1,461 (71.5) | 1,538 (73.5) | 1,505 (73.6) | 1,434 (71.2) | 1,447 (74.5) | 1,402 (72.7) | 1,360 (73.7) | 1,286 (73.0) | 1,311 (72.0) | 1,218 (72.1) | 1,281 (72.1) | 0.15 |
| ≥1 comorbidity | ≥65 | 701 (65.8) | 946 (65.4) | 1,037 (66.7) | 1,012 (63.6) | 867 (55.5) | 931 (58.5) | 931 (60.1) | 1,046 (66.4) | 1,143 (67.5) | 1,146 (67.9) | 1,164 (70.5) | 1,311 (70.4) | 1,303 (69.7) | 1,463 (72.0) | 1,506 (72.0) | <0.001 |

Unless otherwise indicated, data are expressed as number (%) of patients.

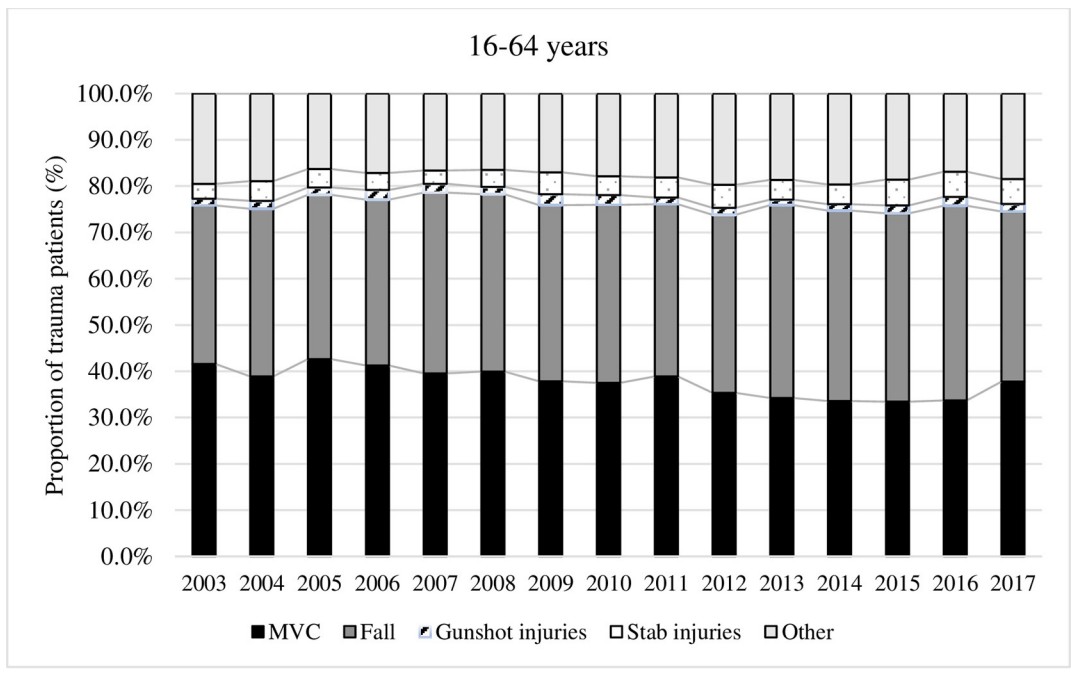

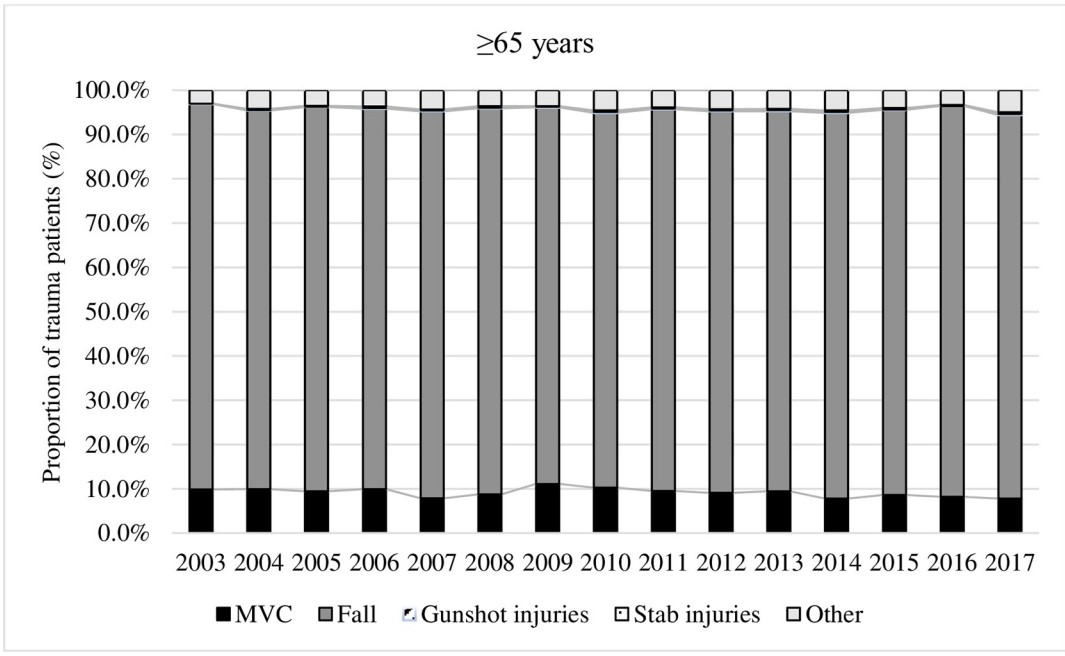

**Fig 2. Evolution of injury mechanisms by age group.** Falls include: fall from own height and fall from higher that own height, MVC: Motor Vehicle Collision.

p-trend<0.001) and lower extremities injuries significantly decreased (-24.2% and -5.7%, p-trend<0.001) over the years. The proportion of severe head trauma significantly increased by 8.7% between 2003 and 2017 (p-trend = 0.01) in older adults but remained stable among younger adults (p-trend = 0.08). While a 5.6% increase in severe abdominal injuries was observed among younger adults (p-trend<0.001), no significant change was found in older adults (p-trend = 0.9).

Table 2. Comparison of injury pattern (AIS≥1) between ≥ 65 years and 16–64 years patients over the years.

| | | 2003 | 2004 | 2005 | 2006 | 2007 | 2008 | 2009 | 2010 | 2011 | 2012 | 2013 | 2014 | 2015 | 2016 | 2017 | p-trend |
|---|---|---|---|---|---|---|---|---|---|---|---|---|---|---|---|---|---|
| | | n = 2,545 | n = 3,378 | n = 3,632 | n = 3,642 | n = 3,604 | n = 3,685 | n = 3,594 | n = 3,589 | n = 3,637 | n = 3,617 | n = 3,495 | n = 3,623 | n = 3,692 | n = 3,723 | n = 3,868 | |
| Head | ≥ 65 | 261 (24.5) | 387 (26.7) | 429 (27.6) | 427 (26.9) | 433 (27.7) | 483 (30.3) | 536 (34.6) | 557 (35.4) | 660 (39) | 635 (37.6) | 619 (37.5) | 694 (37.3) | 694 (37.1) | 752 (37) | 779 (37.3) | <0.001 |
| | 16–64 | 652 (44.1) | 819 (42.4) | 887 (42.7) | 920 (44.8) | 838 (41) | 927 (44.3) | 880 (43) | 881 (43.7) | 883 (45.4) | 811 (42.1) | 783 (42.4) | 749 (42.5) | 804 (44.1) | 732 (43.3) | 846 (47.6) | 0.06 |
| Face | ≥ 65 | 160 (49.7) | 181 (42.6) | 234 (45.5) | 256 (42.2) | 217 (39.8) | 227 (44.1) | 271 (41.1) | 293 (39.2) | 346 (38.2) | 296 (37.8) | 294 (34.7) | 373 (30.7) | 353 (30.8) | 403 (28.6) | 465 (30.1) | <0.001 |
| | 16–64 | 529 (35.7) | 617 (42.4) | 707 (42.7) | 671 (44.8) | 621 (41.0) | 702 (44.3) | 636 (43.0) | 618 (43.7) | 647 (45.4) | 638 (42.1) | 573 (42.4) | 571 (42.5) | 576 (44.1) | 582 (43.2) | 629 (47.6) | 0.7 |
| Thorax | ≥ 65 | 97 (9.1) | 155 (10.7) | 195 (12.5) | 198 (12.5) | 146 (9.3) | 208 (13.1) | 235 (15.2) | 246 (15.6) | 285 (16.8) | 293 (17.3) | 318 (19.3) | 363 (19.5) | 350 (18.7) | 394 (19.4) | 410 (19.6) | <0.001 |
| | 16–64 | 382 (25.8) | 490 (25.4) | 580 (27.9) | 587 (27.9) | 568 (27.8) | 612 (29.2) | 603 (29.5) | 633 (31.4) | 604 (31.1) | 582 (30.2) | 551 (29.9) | 558 (31.7) | 600 (32.9) | 585 (34.6) | 705 (39.7) | <0.001 |
| Spine | ≥ 65 | 77 (7.2) | 121 (8.4) | 154 (9.9) | 161 (10.1) | 182 (11.7) | 208 (13.1) | 208 (13.4) | 239 (15.2) | 258 (15.2) | 286 (16.9) | 284 (17.2) | 321 (17.2) | 323 (17.3) | 357 (17.6) | 339 (16.2) | <0.001 |
| | 16–64 | 332 (22.4) | 408 (21.1) | 488 (23.5) | 492 (24.0) | 510 (25.0) | 488 (23.3) | 511 (25.0) | 505 (25.1) | 524 (27.0) | 528 (27.4) | 462 (25.0) | 417 (23.7) | 446 (24.5) | 435 (25.7) | 525 (29.5) | <0.001 |
| Abdomen | ≥ 65 | 43 (4.0) | 51 (3.5) | 54 (3.5) | 49 (3.1) | 51 (3.3) | 57 (3.6) | 53 (3.4) | 55 (3.5) | 74 (4.4) | 75 (4.4) | 87 (5.3) | 98 (5.3) | 102 (5.5) | 120 (5.9) | 115 (5.5) | <0.001 |
| | 16–64 | 258 (17.4) | 318 (16.5) | 381 (18.3) | 338 (16.5) | 370 (18.1) | 343 (16.4) | 317 (15.5) | 308 (15.3) | 344 (17.7) | 313 (16.2) | 313 (17) | 288 (16.3) | 325 (17.8) | 330 (19.5) | 376 (21.2) | 0.01 |
| Upper extremities | ≥ 65 | 263 (24.7) | 332 (22.9) | 391 (25.2) | 405 (25.5) | 365 (23.4) | 426 (26.8) | 396 (25.6) | 405 (25.7) | 485 (28.6) | 427 (25.3) | 459 (27.8) | 518 (27.8) | 552 (29.5) | 606 (29.8) | 641 (30.7) | <0.001 |
| | 16–64 | 258 (17.4) | 318 (16.5) | 381 (18.3) | 338 (16.5) | 370 (18.1) | 343 (16.4) | 317 (15.5) | 308 (15.3) | 344 (17.7) | 313 (16.2) | 313 (17.0) | 288 (16.3) | 325 (17.8) | 330 (19.5) | 376 (21.2) | <0.001 |
| Lower extremities | ≥ 65 | 751 (70.5) | 968 (66.9) | 1,043 (67.1) | 1,055 (66.4) | 960 (61.5) | 944 (59.3) | 845 (54.6) | 821 (52.1) | 882 (52.1) | 843 (49.9) | 869 (52.7) | 976 (52.4) | 953 (51.0) | 1,061 (52.2) | 1,141 (54.6) | <0.001 |
| | 16–64 | 752 (50.8) | 962 (49.8) | 1,085 (52.2) | 1,024 (49.9) | 1,047 (51.3) | 1,091 (52.1) | 1,033 (50.5) | 970 (48.2) | 927 (47.7) | 917 (47.6) | 876 (47.5) | 851 (48.3) | 893 (49.0) | 854 (50.5) | 910 (51.2) | 0.03 |

AIS: abbreviated injury scale

Unless otherwise indicated, data are expressed as number (%) of patients.

Table 3. Comparison of severe (AIS≥3) injury pattern between ≥ 65 years and 16–64 years patients over the years.

| | | 2003 n = 2,545 | 2004 n = 3,378 | 2005 n = 3,632 | 2006 n = 3,642 | 2007 n = 3,604 | 2008 n = 3,685 | 2009 n = 3,594 | 2010 n = 3,589 | 2011 n = 3,637 | 2012 n = 3,617 | 2013 n = 3,495 | 2014 n = 3,623 | 2015 n = 3,692 | 2016 n = 3,723 | 2017 n = 3,868 | p-trend |
|---|---|---|---|---|---|---|---|---|---|---|---|---|---|---|---|---|---|
| Head | ≥ 65 | 179 (16.8) | 263 (18.2) | 310 (19.9) | 311 (19.6) | 330 (21.1) | 375 (23.6) | 411 (26.5) | 434 (27.6) | 469 (27.7) | 478 (28.3) | 454 (27.5) | 509 (27.4) | 495 (26.5) | 526 (25.9) | 534 (25.5) | **0.01** |
| | 16–64 | 410 (27.7) | 529 (27.4) | 537 (25.8) | 593 (28.9) | 535 (26.2) | 613 (29.3) | 575 (28.1) | 595 (29.5) | 572 (29.4) | 547 (28.4) | 515 (27.9) | 484 (27.5) | 503 (27.6) | 455 (26.9) | 497 (28.0) | 0.08 |
| Face | ≥ 65 | 0 (0) | 1 (0.1) | 4 (0.3) | 3 (0.2) | 1 (0.1) | 4 (0.3) | 2 (0.1) | 1 (0.1) | 6 (0.4) | 7 (0.4) | 6 (0.4) | 6 (0.3) | 2 (0.1) | 5 (0.2) | 4 (0.2) | 0.3 |
| | 16–64 | 14 (0.9) | 19 (1.0) | 15 (0.7) | 26 (1.3) | 17 (0.8) | 27 (1.3) | 17 (0.8) | 24 (1.2) | 23 (1.2) | 27 (1.4) | 38 (2.1) | 20 (1.1) | 22 (1.2) | 27 (1.6) | 29 (1.6) | **0.01** |
| Thorax | ≥ 65 | 34 (3.2) | 55 (3.8) | 72 (4.6) | 74 (4.7) | 71 (4.5) | 102 (6.4) | 125 (8.1) | 114 (7.2) | 138 (8.1) | 128 (7.6) | 171 (10.4) | 199 (10.7) | 216 (11.6) | 264 (13.0) | 254 (12.1) | **<0.001** |
| | 16–64 | 177 (12.0) | 255 (13.2) | 298 (14.3) | 317 (15.4) | 337 (16.5) | 332 (15.9) | 344 (16.8) | 374 (18.6) | 365 (18.8) | 342 (17.7) | 349 (18.9) | 371 (21.1) | 400 (22.0) | 389 (23.0) | 494 (27.8) | **<0.001** |
| Spine | ≥ 65 | 26 (2.4) | 49 (3.4) | 73 (4.7) | 66 (4.2) | 75 (4.8) | 88 (5.5) | 92 (5.9) | 126 (8) | 129 (7.6) | 138 (8.2) | 134 (8.1) | 163 (8.8) | 154 (8.2) | 164 (8.1) | 155 (7.4) | **0.003** |
| | 16–64 | 118 (8.0) | 189 (9.8) | 227 (10.9) | 215 (10.5) | 220 (10.8) | 201 (9.6) | 213 (10.4) | 236 (11.7) | 230 (11.8) | 223 (11.6) | 215 (11.7) | 188 (10.7) | 200 (11.0) | 198 (11.7) | 232 (13.1) | **0.03** |
| Abdomen | ≥ 65 | 5 (0.5) | 6 (9.4) | 5 (0.3) | 8 (0.5) | 7 (0.4) | 5 (0.3) | 17 (1.1) | 9 (0.6) | 22 (1.3) | 14 (0.8) | 29 (1.8) | 19 (1.0) | 21 (1.1) | 26 (1.3) | 12 (0.6) | 0.9 |
| | 16–64 | 49 (3.3) | 67 (3.5) | 101 (4.9) | 91 (4.4) | 105 (5.1) | 93 (4.4) | 112 (5.5) | 101 (5.0) | 105 (5.4) | 99 (5.1) | 121 (6.6) | 90 (5.1) | 115 (6.3) | 111 (6.6) | 159 (8.9) | **<0.001** |
| Upper extremities | ≥ 65 | 38 (3.6) | 59 (4.1) | 48 (3.1) | 52 (3.3) | 53 (3.4) | 69 (4.3) | 61 (3.9) | 52 (3.3) | 57 (3.4) | 50 (3.0) | 19 (1.2) | 15 (0.8) | 13 (0.7) | 12 (0.6) | 16 (0.8) | **<0.001** |
| | 16–64 | 91 (6.1) | 138 (7.1) | 132 (6.4) | 128 (6.2) | 113 (5.5) | 129 (6.2) | 115 (5.6) | 103 (5.1) | 105 (5.4) | 106 (5.5) | 48 (2.6) | 35 (2.0) | 55 (3.0) | 37 (2.2) | 43 (2.4) | **<0.001** |
| Lower extremities | ≥ 65 | 563 (52.9) | 729 (50.4) | 754 (48.5) | 749 (47.1) | 679 (43.5) | 619 (38.9) | 562 (36.3) | 536 (34.0) | 560 (33.1) | 525 (31.1) | 464 (28.1) | 509 (27.4) | 500 (26.7) | 541 (26.6) | 599 (28.7) | **<0.001** |
| | 16–64 | 347 (23.5) | 453 (23.5) | 550 (26.4) | 461 (22.5) | 455 (22.3) | 491 (23.5) | 489 (23.9) | 471 (23.4) | 462 (23.8) | 448 (23.2) | 320 (17.3) | 281 (16.0) | 290 (15.9) | 293 (17.3) | 316 (17.8) | **<0.001** |

AIS: abbreviated injury scale

Data are expressed as number (%) of patients.

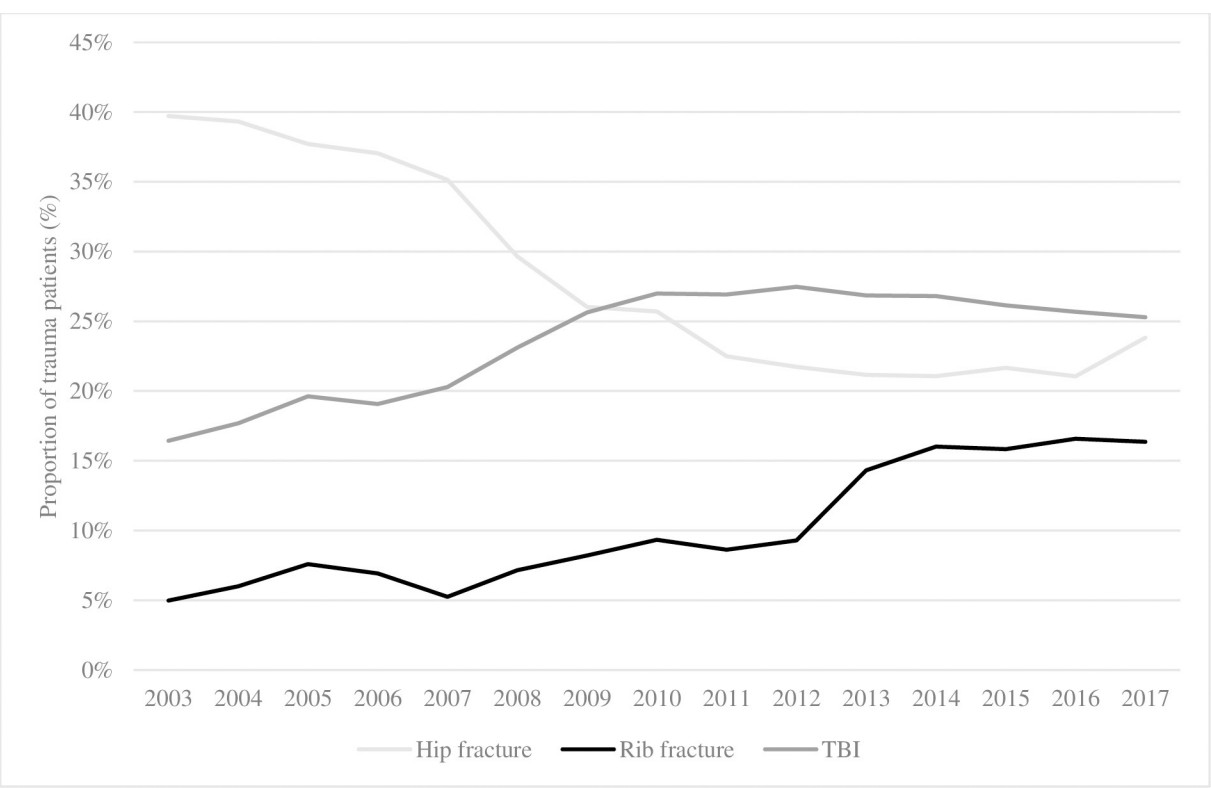

**Fig 3. Proportion of older patients presenting with hip fracture, rib(s) fracture or traumatic brain injury.** TBI: traumatic brain injury.

Regarding more specific injuries, we found that the proportion of older adults with hip fractures decreased by 15.9% over time, but rib fracture(s) and TBI increased by 11.4% and 8.9%, respectively (Fig 3).

## Injury severity and mortality

The proportion of severely injured (ISS>12) older adults significantly increased over the years (p-trend <0.001), and almost doubled from 17.6% (n = 187) in 2003 to 32.3% (n = 675) in 2017. Nonetheless, a 1.0% decrease in mortality was observed in this population between 2003 and 2017, but this was not associated with a significant change in mortality over the years (p-trend = 0.335). A significant increase was also observed in the proportion of very severely injured (ISS≥25) older adults admitted between 2003 and 2017: from 7% (n = 75) to 15.7% (n = 328, p-trend <0.001). Mortality in this group significantly decreased over the years (p-trend <0.001).

Similarly, the proportion of patients aged 16–64 years with an ISS>12 increased by 12.1% between 2003 and 2017 (p-trend <0.001) while mortality significantly decreased along this period (p-trend<0.001). The proportion of younger patients with an ISS>25 also significantly increased by 5.1% between 2003 and 2017 (p-trend = 0.0055), while mortality significantly decreased over the year (p-trend = 0.005; Fig 4).

## In-hospital LOS, complications, and discharge destinations

The mean hospital LOS ranged from 15.7 to 19.3 days in older adults and was higher than their younger counterparts (from 12 to 14.2 days). The absolute number of annual bed-days in

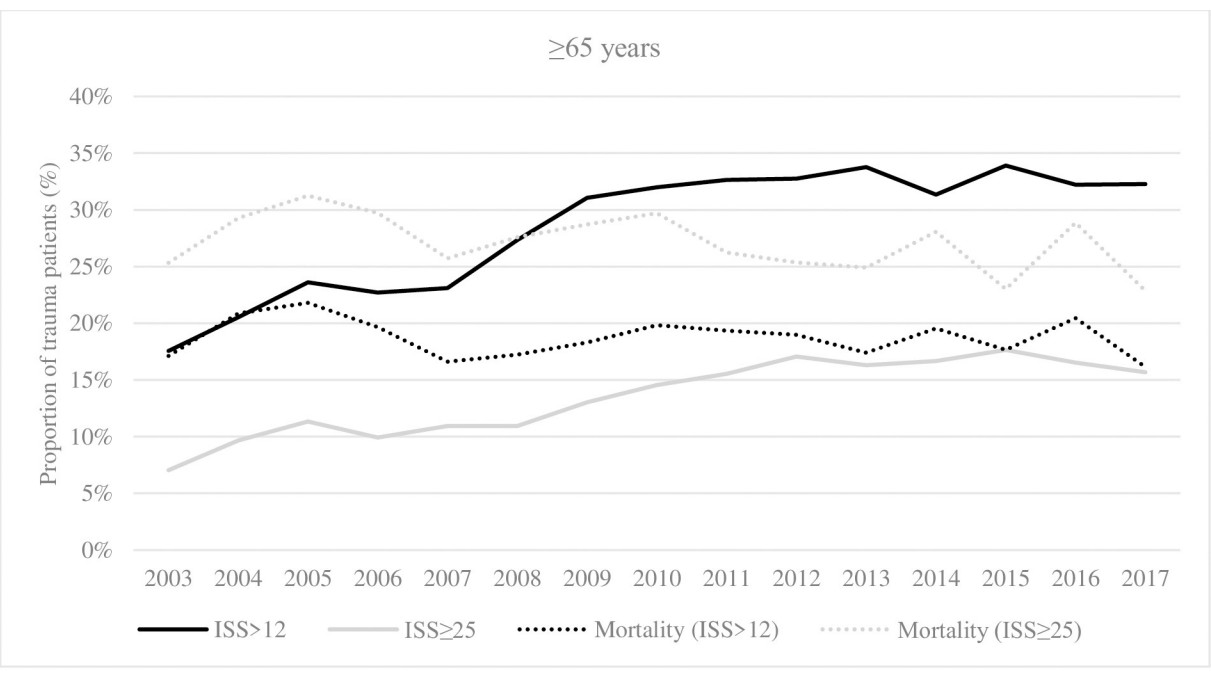

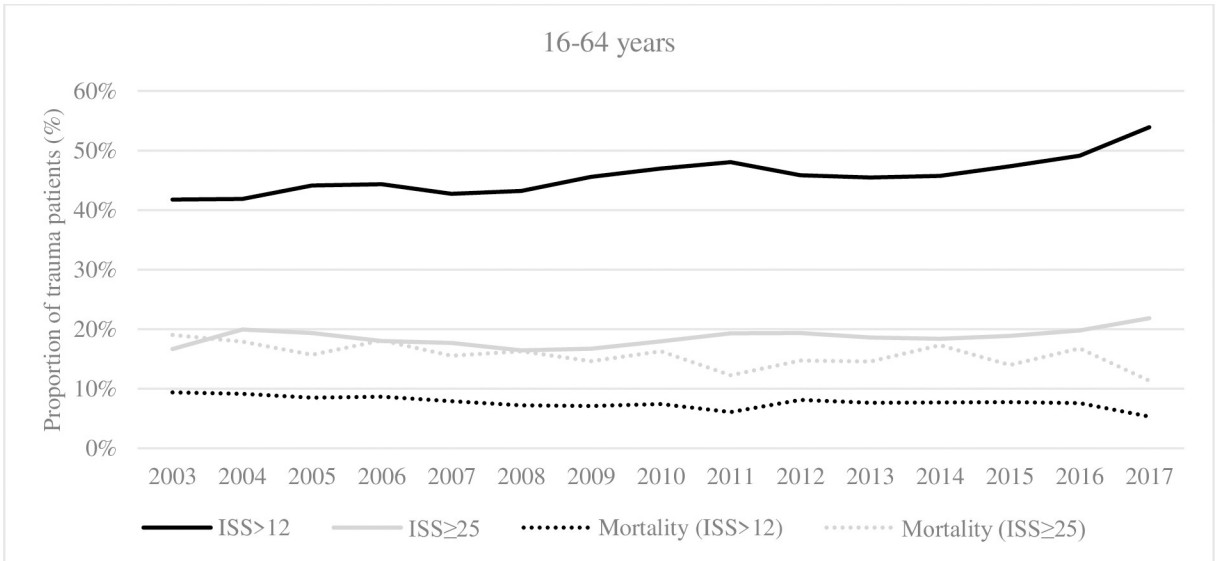

**Fig 4. Evolution of severe trauma patients and their mortality by age group.** ISS: Injury Severity Score.

older adults increased in non-ICU and ICU (+15,004 and +1,437, respectively). Since 2008, the bed-days consumption of older adults has been higher in the non-ICU ward than younger patients (Fig 5).

Older adults more frequently experienced in-hospital complications compared to their younger counterparts and this increased over time in both groups (17.4% to 22.7% and 9.8% to 15.3%, respectively, p-trend <0.001, Fig 6). Further details on the evolution of the type of complications are presented in S1 Table.

Discharge destinations are reported in Fig 7. The proportion of older adults discharged home slightly increased from 32.8% to 36.4%. Transfer to another health care facility increased

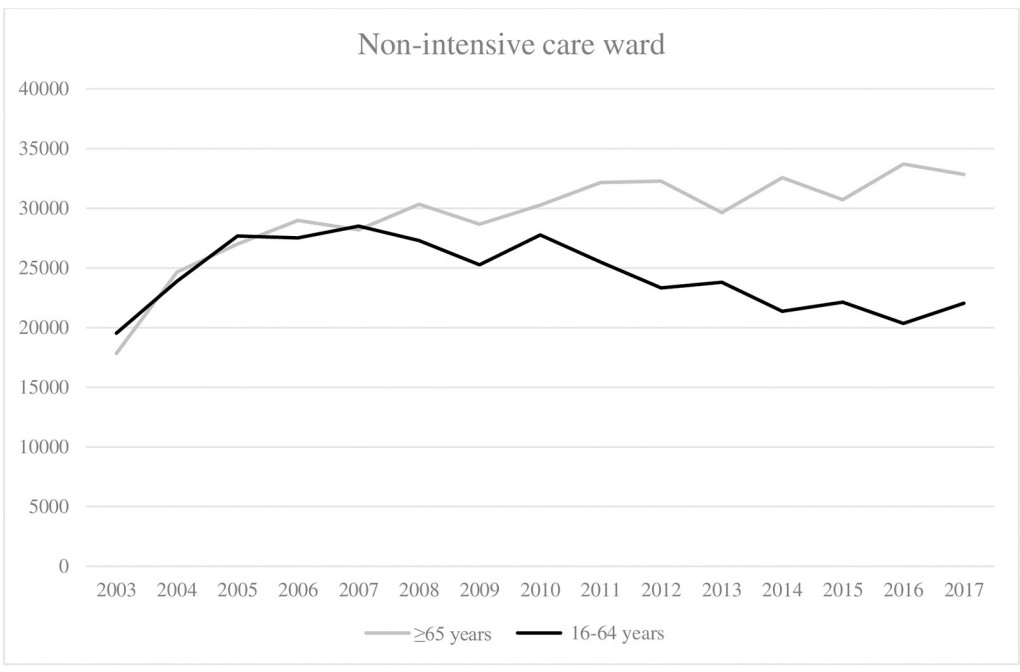

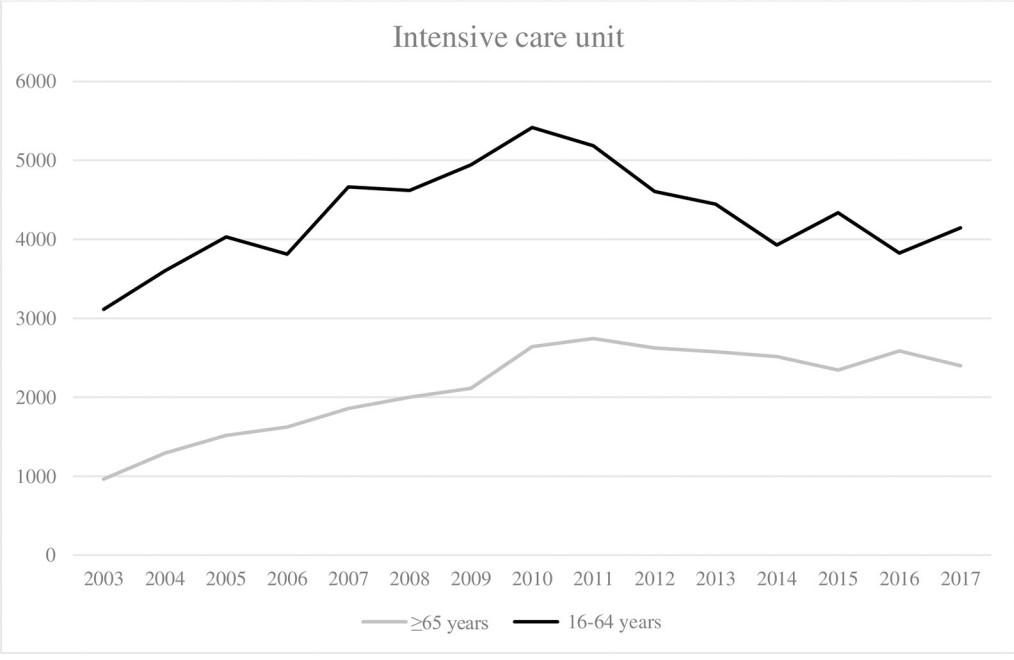

**Fig 5. Absolute number of bed-days over the years.**

from 8.9% to 17.3%. Conversely, the proportion of patients admitted to a rehabilitation center (from 37.3% to 27.2%), or long-term facility care (from 13.2% to 10%) decreased.

## Discussion

Our study showed that the median age of Canadian patients admitted to Québec province level-I trauma centers constantly increased and that their injury pattern has mutated over the

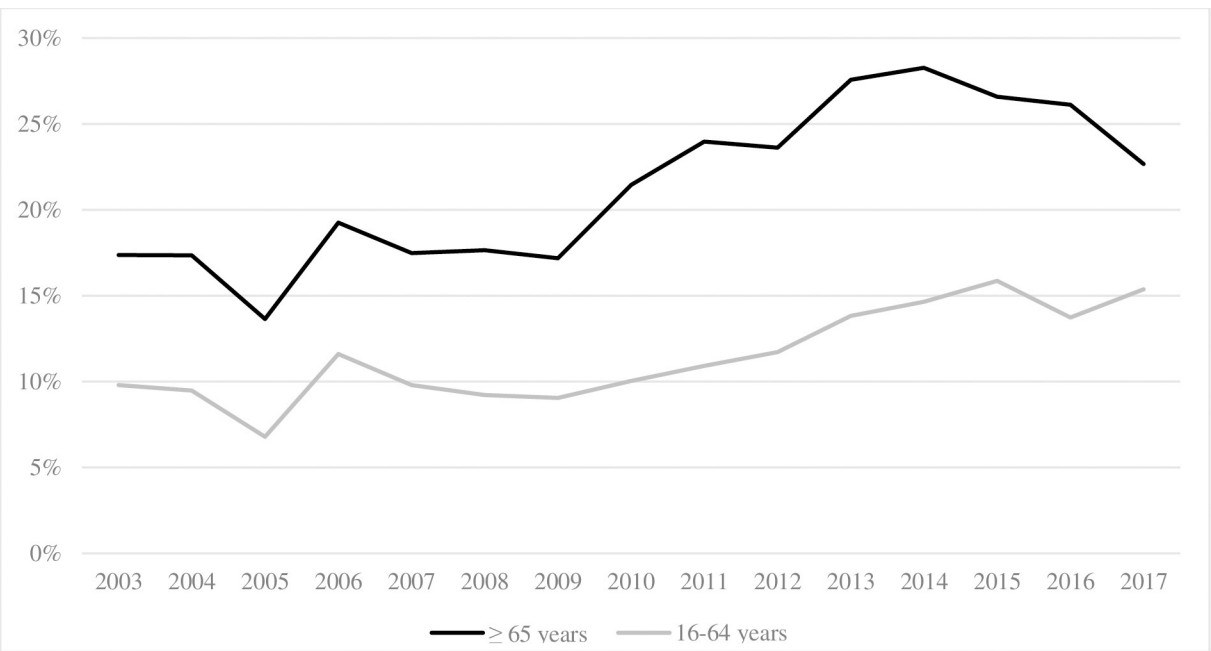

**Fig 6. Evolution of in-hospital complications.**

years. The severity of older patients' injuries increased, but we showed a decrease in mortality in both age groups, even in the most severe trauma patients.

Older trauma patients are now more numerous than younger adults, which is consistent with reported international information [2, 3, 21–27]. Our results not only confirm previous findings that fall are the most common injury mechanism among older adults [28, 29] but also show that they are very prevalent in younger adults as they have interchangeably been the leading cause of trauma with MVC in that population. The proportion of older patients who sustained severe head, thorax, and spine injuries increased considerably over time. An Australian study similarly reported a significant increase in the population-adjusted incidence of thoracic injury (8%/year), with the most significant increase being in patients aged ≥85 years (14%/year) [30]. An increasing number of older adults with spine injuries has been described elsewhere as well [29]. A possible explanation for this phenomenon is the increased sensitivity and specificity of computed tomography (CT) for spine injuries over conventional radiography [31–33]. Conversely, we observed a decrease in the proportion of older adults with an upper or lower extremities injury. This was also noted by Kalbas et al. However, their results were possibly related to the decreasing incidence of high-energy traumas (i.e. road traffic accidents), which was not the case in our population [31]. The decreasing incidence of hip fractures, one of the most frequent lower extremity injuries in older adults [34], may partially explain our findings. Other authors recently supported this and could be attributed to improved osteoporosis treatment and healthier lifestyle choices [35]. The evolution of injury patterns may also be attributed to improvements in diagnostic radiology [26] combined with the recent democratization of whole-body computed tomography (WBCT), which became standard practice in many centers in the last two decades [29, 36].

The proportion of severe trauma patients considerably increased, especially among older adults. This has also been observed in the English Trauma Audit Research Network database [37] and another large US cohort study [22]. Despite the increased severity and comorbidity burden, we noted a slight decrease in mortality in both age groups. Dinh et al. reported a 2.2%

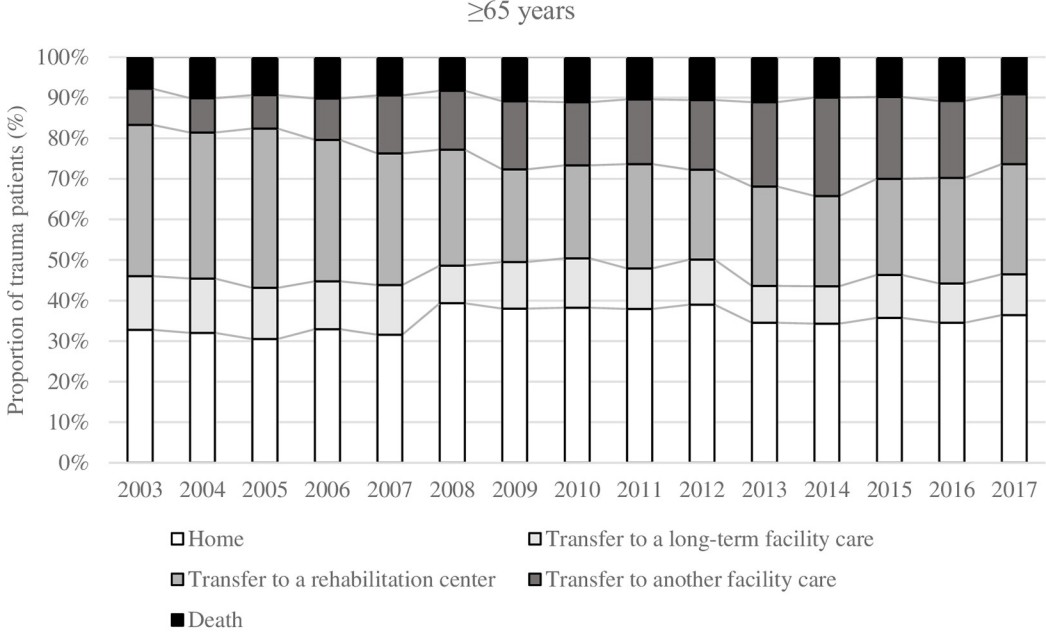

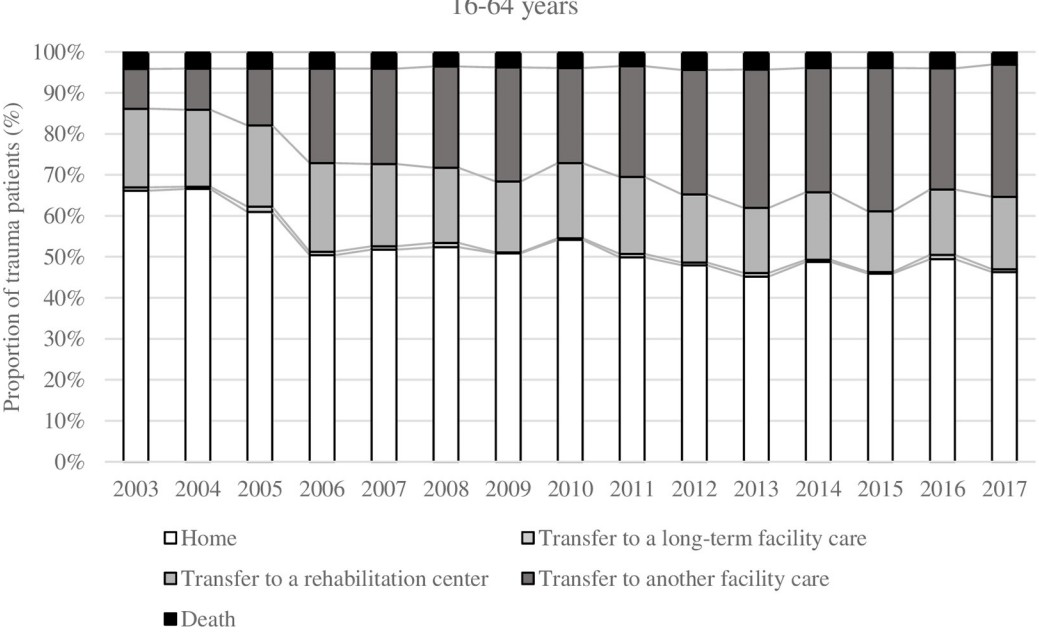

**Fig 7. Hospital discharge destinations over the years.**

standardized mortality drop per year in older adults. The authors explained this was related to a decline in pedestrian injuries, who commonly sustain more severe injuries and have higher mortality rates but also to improved prehospital and acute care [2]. We also found that older trauma patients experienced longer in-hospital LOS, which is in line with the results reported by other authors [38–40]. This could be related to the higher incidence of complications in this population [41–44]. However, we cannot rule out that these complications may also be the

consequence of longer LOS. In addition, we noted a significant increase in bed-day consumption per year among older adults. Older patients were more frequently admitted to rehabilitation centers or short/long-term care facilities than younger adults. All these are essential drivers of increased healthcare costs.

Our large multicenter cohort includes consecutive trauma patients who consulted or were transferred to any of the province's three Level-I trauma centers between 2003 and 2017, which is a non-negligible strength. Furthermore, the little missing data mainly pertained to comorbidities (S2 Table). This epidemiological study is a reliable overview of the Canadian trauma population as a whole, as other studies have focussed on specific trauma mechanism [45], or population [25, 28, 46]. In addition, studies exploring temporal changes in injury pattern and severity are sparse [26, 28, 29]. At last, because advanced age has been associated with undertriage [6–8] (inaccurate triage that results in a patient who requires higher-level care not being directly transported to a Level-I or Level-II trauma center), we also included secondarily transferred patients.

Health policymakers and trauma care providers should consider these results to offer high-quality trauma care, a more dedicated care pathway, and senior-friendly triage tools. Concurrently, interventions shown to reduce falls must be actively deployed.

The changing characteristics of Canadian trauma patients and the economic impact of these changes need to be further studied. The differences between the old (65–74 years), older (75–84 years), and oldest ($\geq$ 85 years) patients may also need further investigation, as some differences have previously been highlighted [47, 48] but have yet to be investigated in a Canadian population.

## Limitations

This study has limitations. First, only patients admitted or transferred to a level-I trauma center were included, which may have overestimated trauma severity and mortality. Nevertheless, we focused on patients with significant trauma since minor injuries are often discharged home and do not generate the same burden in terms of mortality and healthcare costs. Some of the increases in injuries to specific body regions may be explained by the increased use of CT/WBCT. However, this may impact older and younger adults alike. The incidence of some specific injuries may have been underestimated in the early 2000s, and this was a factor we could not account for. Finally, because of our study's retrospective design, some variables, such as complications, may have been underestimated in cases where they were not reported in the patients' charts.

## Conclusion

Since 2014, older adults have represented the majority of admissions in Level-I trauma centers in the province of Québec, Canada. Meanwhile, the injury pattern of these patients has deeply evolved, and the ageing trauma population is now more severely injured. Yet, we showed a decrease in mortality, even in the most severe trauma patients. Bed-day consumption has greatly increased in this population, and a medico-economic evaluation is needed to assess the burden related to these findings. In addition, offering high-level trauma care, a dedicated pathway of care, and senior-friendly triage tools are a clear priority.

## Supporting information

**S1 Table. Evolution of complications over the years.** Respiratory complications include acute respiratory distress syndrome; aspiration pneumonia; acute respiratory failure. Cardiovascular complications include cardiac arrest; myocardial infarction; pulmonary embolism;

deep vein thrombosis; stroke. Surgery-related complications include postoperative haemorrhagic shock; abdominal compartment syndrome; anastomotic leak; evisceration/dehiscence. Infectious complications include: C.difficile colitis; catheter-related bloodstream infection; sepsis/severe sepsis/shock; wound infection; nosocomial pneumonia; osteomyelitis. Other complications include: acute kidney failure; decubitus ulcers; nonunion fracture; delirium; coagulopathy.
(DOCX)

**S2 Table. Missing data over the years.**
(DOCX)

## Acknowledgments

The authors would like to thank the Emergency Physicians from the Hôpital de l'Enfant-Jésus.

## Author Contributions

**Conceptualization:** Axel Benhamed, Brice Batomen, Francis Bernard, Lynne Moore, Marcel Émond.

**Data curation:** Chartelin Jean Isaac.

**Formal analysis:** Brice Batomen, Chartelin Jean Isaac.

**Funding acquisition:** Marcel Émond.

**Investigation:** Brice Batomen, Marcel Émond.

**Methodology:** Brice Batomen, Marcel Émond.

**Project administration:** Valérie Boucher.

**Resources:** Valérie Boucher, Marcel Émond.

**Software:** Chartelin Jean Isaac.

**Supervision:** Marcel Émond.

**Validation:** Marcel Émond.

**Visualization:** Axel Benhamed, Chartelin Jean Isaac, Marcel Émond.

**Writing – original draft:** Axel Benhamed.

**Writing – review & editing:** Brice Batomen, Valérie Boucher, Krishan Yadav, Éric Mercier, Chartelin Jean Isaac, Mélanie Bérubé, Francis Bernard, Jean- Marc Chauny, Lynne Moore, Marie Josée Sirois, Karim Tazarourte, Amaury Gossiome, Marcel Émond.

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
