## [Decision Letter · Decision Letter 0]

11 Oct 2022

PONE-D-22-22014Epidemiology, injury pattern and outcome of older trauma patients: a 15-year study of level-I trauma centersPLOS ONE

Dear Dr. Émond,

Thank you for submitting your manuscript to PLOS ONE. After careful consideration, we feel that it has merit but does not fully meet PLOS ONE’s publication criteria as it currently stands. Therefore, we invite you to submit a revised version of the manuscript that addresses the points raised during the review process.

We look forward to receiving your revised manuscript.

Kind regards,

Yusuke Tsutsumi

Academic Editor

PLOS ONE

Journal Requirements:

 "AB received a scholarship from the Fondation du CHU de Québec (no grant number). This project was funded by the Fonds de recherche du Québec – Santé ( #33239). These funding agencies were in no way involved in the design, data collection and analyses of this study"

Reviewers' comments:

Reviewer's Responses to Questions

**Comments to the Author**

1. Is the manuscript technically sound, and do the data support the conclusions?

Reviewer #1: Yes

Reviewer #2: Partly

2. Has the statistical analysis been performed appropriately and rigorously? 

Reviewer #1: Yes

Reviewer #2: Yes

3. Have the authors made all data underlying the findings in their manuscript fully available?

Reviewer #1: Yes

Reviewer #2: Yes

4. Is the manuscript presented in an intelligible fashion and written in standard English?

Reviewer #1: Yes

Reviewer #2: Yes

5. Review Comments to the Author

Reviewer #1: Dear authors

Thank you for giving me the opportunity to review this interesting manuscript. This theme is very important. Your manuscript was well written. So I have only one question.

Table 1; Why can you say there is significant (or non significant) change in Table1? Because you did not use statistical test in table 1.

Reviewer #2: 1． General comments

This descriptive study evaluated temporal changes in incidence, demographic and trauma characteristics, injury patterns, in-hospital admission, complications, and outcome of older trauma patients at three Level 1 trauma centers in Quebec, Canada, from March 2003 to December 2017. In particular, the study showed an increase in the proportion of elderly patients aged 65 years and older, an increase in the proportion of elderly patients in the severe trauma population, and an increase in the number of bed days used by elderly patients.

　The study showed changes in injury trajectory, site of injury, and severity over 15 years at three Level 1 trauma centers that admit the most severely traumatized patients. The study also showed that the burden on Level 1 trauma centers has increased, particularly with an increase in the proportion of patients aged 65 years and older, as well as the number of hospital days and complications. The results may contribute to efficiently allocating healthcare resources for hospital personnel and health policymakers. It may also lead to further descriptive studies that are more complete, such as the creation of a database to capture trauma patients across Canada. However, the study has several concerns for improvement at this time.

2． Specific comments

a)Major

1)In the comparison between 2003 and 2017, it is stated that the mortality rate of the elderly decreased. However, referring to Fig. 4, there seems to be almost no change in the mortality over time in either group with ISS>12 or ISS≥25 (the authors might do a trend test). Comparing only the first and the last year and stating that mortality decreased is not well-founded, and it is recommended that the summary and conclusions be written in a less toned-down manner.

2)The percentage of severe lower extremity trauma in Tab. 3 for those 65 and older decreased significantly after 2012 and 2013. The discussion cites the improvement of osteoporosis and the contribution of a healthy lifestyle as the cause. However, it is unlikely to cause a marked decrease in severe trauma to the lower extremities after one year; the number of hospitalized patients aged 65 years or older from Fig. 1 increased sharply around 2014. It is recommended to consider other factors. For instance, was there a change in the local medical response, such as transferring severe lower extremity trauma patients to lower-level trauma centers? Is there a possibility of misclassification due to diagnostic methods and AIS calculations?

3)Many observations included in complications are listed, but the aggregation method of these observations is unclear. Although the percentage of complications is shown in Fig. 6, a description of specific complications and characteristics could provide readers with helpful information for actual patient management. Although the amount of information would be enormous, consideration could be given to presenting only a few years or a few complications.

b)Minor

1)I recommend that the reasons for excluding deaths before or on arrival at the hospitals be described.

2)It may be helpful to specify the number of deaths before or on arrival at the hospitals. In that case, it is unlikely that a detailed search for the injury site would have been conducted, but if the age, rough cause of death, and trajectory of injury could be determined, it might be a helpful description. This is because this study aims to describe the epidemiology of patients transported to the trauma center, not to evaluate the treatment outcomes of the Level 1 trauma center.

6. PLOS authors have the option to publish the peer review history of their article (what does this mean?). If published, this will include your full peer review and any attached files.

Reviewer #1: No

Reviewer #2: **Yes: **Hashimoto K

---

## [Author Response · Author response to Decision Letter 0]

7 Dec 2022

See the point by point response in the attached response letter.

---

## [Decision Letter · Decision Letter 1]

28 Dec 2022

Epidemiology, injury pattern and outcome of older trauma patients: a 15-year study of level-I trauma centers

PONE-D-22-22014R1

Dear Dr. Émond,

We’re pleased to inform you that your manuscript has been judged scientifically suitable for publication and will be formally accepted for publication once it meets all outstanding technical requirements.

Kind regards,

Yusuke Tsutsumi

Academic Editor

PLOS ONE

Additional Editor Comments (optional):

Reviewers' comments:

Reviewer's Responses to Questions

**Comments to the Author**

1. If the authors have adequately addressed your comments raised in a previous round of review and you feel that this manuscript is now acceptable for publication, you may indicate that here to bypass the “Comments to the Author” section, enter your conflict of interest statement in the “Confidential to Editor” section, and submit your "Accept" recommendation.

Reviewer #1: All comments have been addressed

Reviewer #2: (No Response)

2. Is the manuscript technically sound, and do the data support the conclusions?

Reviewer #1: Yes

Reviewer #2: (No Response)

3. Has the statistical analysis been performed appropriately and rigorously? 

Reviewer #1: Yes

Reviewer #2: (No Response)

4. Have the authors made all data underlying the findings in their manuscript fully available?

Reviewer #1: Yes

Reviewer #2: (No Response)

5. Is the manuscript presented in an intelligible fashion and written in standard English?

Reviewer #1: Yes

Reviewer #2: (No Response)

6. Review Comments to the Author

Reviewer #1: (No Response)

Reviewer #2: (No Response)

7. PLOS authors have the option to publish the peer review history of their article (what does this mean?). If published, this will include your full peer review and any attached files.

Reviewer #1: No

Reviewer #2: No

---

## [Editor Report · Acceptance letter]

20 Jan 2023

PONE-D-22-22014R1 

Epidemiology, injury pattern and outcome of older trauma patients: a 15-year study of level-I trauma centers 

Dear Dr. Émond:

I'm pleased to inform you that your manuscript has been deemed suitable for publication in PLOS ONE. Congratulations! Your manuscript is now with our production department. 

Kind regards, 

on behalf of

Dr. Yusuke Tsutsumi 

Academic Editor

PLOS ONE